# MKGL: Mastery of a Three-Word Language

**Lingbing Guo**[1,2], **Zhongpu Bo**[3], **Zhuo Chen**[1,2], **Yichi Zhang**[1,2], **Jiaoyan Chen**[4],
**Yarong Lan**[1,2], **Mengshu Sun**[3], **Zhiqiang Zhang**[3], **Yangyifei Luo**[5], **Qian Li**[6],
**Qiang Zhang**[1,2], **Wen Zhang**[7,2*] **and Huajun Chen**[1,2*]

[1]College of Computer Science and Technology, Zhejiang University
[2]ZJU-Ant Group Joint Lab of Knowledge Graph
[3]Ant Group
[4]Department of Computer Science, The University of Manchester
[5]School of Computer Science and Engineering, Beihang University
[6]School of Computer Science, Beijing University of Posts and Telecommunications
[7]School of Software Technology, Zhejiang University

## Abstract

Large language models (LLMs) have significantly advanced performance across
a spectrum of natural language processing (NLP) tasks. Yet, their application to
knowledge graphs (KGs), which describe facts in the form of triplets and allow
minimal hallucinations, remains an underexplored frontier. In this paper, we
investigate the integration of LLMs with KGs by introducing a specialized KG
Language (KGL), where a sentence precisely consists of an entity noun, a relation
verb, and ends with another entity noun. Despite KGL's unfamiliar vocabulary to
the LLM, we facilitate its learning through a tailored dictionary and illustrative
sentences, and enhance context understanding via real-time KG context retrieval
and KGL token embedding augmentation. Our results reveal that LLMs can
achieve fluency in KGL, drastically reducing errors compared to conventional KG
embedding methods on KG completion. Furthermore, our enhanced LLM shows
exceptional competence in generating accurate three-word sentences from an initial
entity and interpreting new unseen terms out of KGs.

## 1 Introduction

Knowledge graphs (KGs) are important resources for many data-driven applications, offering structured repositories of factual information that empower a variety of intelligent tasks [1, 2]. Yet, the strides made through the rapid advancement of large language models (LLMs) have challenged the conventional reliance on KGs. Nonetheless, LLMs are often critiqued for their susceptibility to generating factually incorrect or nonsensical outputs–a phenomenon known as the "hallucination problem" [3, 4]. Many recent studies propose to resort KGs to mitigate this problem [5–8].

In this paper, we investigate the capacity of LLMs to assimilate and generate knowledge graph facts proficiently. For example, the natural language sentence, "Wendee Lee is an actor in Mighty Morphin Power Rangers," translates into a KG triplet format as *(Wendee Lee, actor of, Mighty Morphin Power Rangers)*. It is worth noting that, English names such as *Wendee Lee* and *Mighty Morphin Power Rangers*, while can serve as identifiers for entities, are perceived as atomic elements within the KG framework. They are indivisible and distinct from their constituent words or characters.

When the LLMs interpret these text identifiers as mere sequences of tokens, they risk producing output that misrepresents entities or relations, therefore compromising the integrity of KG-based tasks. Consequently, existing research that integrates LLMs with KGs tends to limit its scope to relatively straightforward tasks. Examples of these limitations include validating the correctness of

---

*Correspondence to: {zhang.wen, huajunsir}@zju.edu.cn

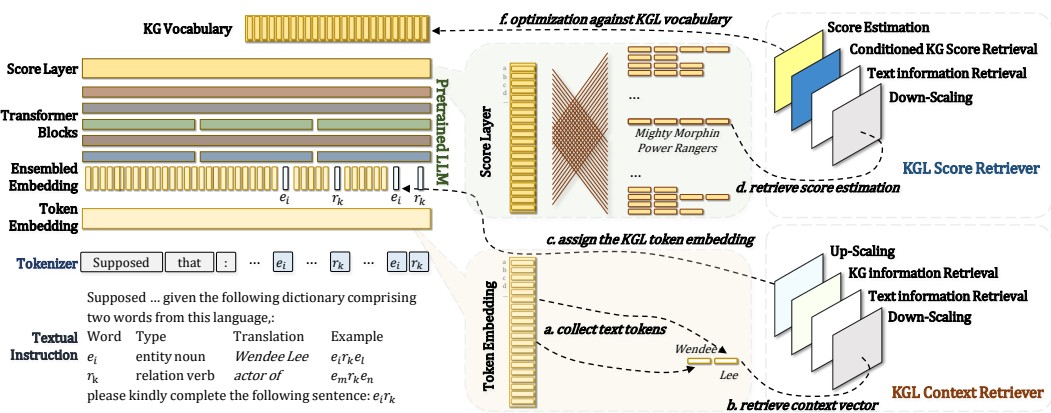

Figure 1: A workflow of MKGL (from bottom to top). The instruction to the LLM includes a dictionary exemplifying the entity $e_i$ and relation $r_k$. The task is to construct new KG sentences initialized with $e_i r_k$. The tokenizer first tokenizes the input text, where the entities and relations are represented as special tokens out of the original vocabulary. (a) To process these special tokens, MKGL collects the embeddings of their constituting text tokens; (b) Then, a retriever performs a 4-step process to aggregate textual and relational information into KGL token embeddings. The first and the last steps are LoRA-like down-scaling and up-scaling operations [12]; (c) The output is assigned as the embeddings of these special KGL tokens; (d) Similar to the context retriever, we design a score retriever to retriever the score information. (f) The output is in a form of probability distribution among candidate entities.

fully-formed triplets [9], or picking an appropriate entity from a limited set of options [10]. Given the sheer volume of entities in a KG, such narrow applications fall short in addressing more complicated tasks like KG completion, wherein a model predicts missing components of a provided incomplete triplet, e.g., identifying the unknown tail entities in *(Wendee Lee, actor of, ?)* against thousands of candidates. While these methods may lean on pretrained KG embedding models to narrow down possible candidates, the process remains inefficient.

To transcend the limitations on predictive scope, we propose a novel approach, named *MKGL*, to instruct an LLM in the lexicon of the unique *KG language (KGL)*. KGL sentences are strictly three-word sentences, starting with an entity noun, followed by a relation verb, and ending with another entity noun. The vocabulary of KGL does not immediately resonate with the machines. A common triplet like *(Wendee Lee, actor of, Mighty Morphin Power Rangers)* is encoded abstractly as $e_i r_k e_j$, with $e_i$, $e_j$ symbolizing the entity nouns and $r_j$ denoting the relation verb. For an LLM such as Llama-2 [11], these symbols are entirely alien, absent from its pretraining corpus. Our investigation thus centers on how an LLM can navigate and master this specialized, atomic language of KGs.

As illustrated in Figure 1, to bridge this comprehension gap, we introduce an English-KGL dictionary, and the LLM is supposed to assemble new KG sentences using the provided linguistic building blocks. The basic elements of KGL, while different from our natural language, are familiar to the LLM as they are constructed from the pretrained token embeddings. We leverage a context retriever to retrieve the text information and relational information of a KGL token, which transforms the sequential token embeddings of its name into an embedding vector. Subsequently, we update the LLM token embedding layer with new KGL token embeddings. In the scoring layer, we also employ a KG score retriever to supplement the LLM with extra KG relational information for prediction.

Instructing an LLM in KGL offers three main advantages over prompt-based methods [10, 13] or conventional KG embedding methods [14, 15]: (1) Broadened applicability. KGL tokens originate from textual tokens of an LLM, thus our method does not mandate that all entities be observed during training. (2) End-to-end framework. Unlike recent LLM-based methods that necessitate pre-sorted results from conventional KG embedding methods, our approach can rapidly rank all candidate entities at one-step. (3) High efficiency. The representations of KG tokens are derived from pretrained token embeddings rather than learned from scratch. The proposed KGL context and score retrievers also leverage a LoRA-like adaption. Using Llama-2-7b [11] as the base LLM, the number of training parameters is less than 0.3%.

However, instructing an LLM in KGL also has its limitations, as it demands more computational resources compared with conventional methods. For instance, fine-tuning MKGL (Llama-2-7b) on the FB15k-237 dataset [16] to outperform most conventional methods requires only 1 epoch. Nevertheless, with 8 A100 GPUs, it still takes half an hour, which is comparable to training a TransE model from scratch with a single GPU.

## 2   Related Works

We category the related works into two groups:

**Knowledge Graph Completion**   KG completion can be regarded as a classification problem like many NLP tasks [17–21], such as node classification and semantic role labeling. However, its label space is significantly larger than most NLP tasks. For example, the WN18RR [22] dataset contains over 40,000 different entities, making it impractical to simply feed them all as possible results and let the LLM select one as output. Most conventional KG completion methods are embedding-based methods, including the triplet-based methods [23–27], e.g, TransE [23], ComplEx [25], RotatE [26]; the GNN-based methods [15, 28–32], e.g., DAN [15], CompGCN [28], CoKE [29]; and other neural-based methods [22, 33–35], e.g., ConvE [22] and RSN [34]. Despite differences in their neural methods and input forms, all these methods focus on relational information and are not good at utilizing other types of information such as textual attributes.

**Pretrained Language Models for Knowledge Graphs**   Leveraging Pretrained language models for KG completion has been explored for many years [36]. Some works treat BERT as a GNN model to encode graph features [30, 37], while others consider the textual information of KGs and use pretrained BERT to encode the textual labels of entities and relations  [14, 38–40]. The resulting outputs are regarded as the entity and relation embeddings or their concatenations.

With the rapid advancements in leveraging LLMs for KG completion, recent works have begun designing prompts or instructions to guide LLMs in this task. Initially, the results were not promising [41], as it appeared that even state-of-the-art LLMs without context information could not outperform basic KG embedding models like TransE. However, subsequent works such as KGLlama [42] and KoPA [13] discovered that LLMs might perform better in triplet classification, i.e., estimating the correctness of a given triplet.

More recently, KICGPT [10] has proposed leveraging in-context learning [43, 44] to provide explicit instructions and guide the behavior of LLMs. This involves a triplet-based KG embedding model to generate the initial rankings of the top-k entities, followed by a multi-round interaction with the LLM, providing textual information and triplet demonstrations for the query entity and relation. The LLM should then re-rank the initial list. KICGPT has achieved state-of-the-art results on KG completion tasks. However, its performance not only depends on the LLM and the instructions but also on the pretrained KG embedding model. Additionally, KICGPT cannot be deployed offline due to the demand of commercial LLMs [45]. It also cannot provide embeddings for downstream tasks.

In contrast, the proposed MKGL has an embedding module based on the LLM token embeddings and KG relational information, which overcomes the weaknesses of existing KG embedding methods that cannot provide embeddings for unseen entities. The context information is implicitly encoded into the KGL token embeddings and efficiently captured by the LLM during fine-tuning.

## 3   Mastery of KG Language

In this section, we discuss the details of MKGL. We first introduce the general architecture of an LLM and how to convert a KG triplet into a fine-tuning instruction. Then, we present the details of constructing KGL token embeddings and scores. Finally, we illustrate how to train an MKGL and analyze its complexity.

### 3.1   Preliminaries

We start by a brief introduction to KGs and LLMs.

**Knowledge Graph and Knowledge Graph Language**    Knowledge graphs are conceptualized as directed, multi-relational graphs. We describe a knowledge graph by $\mathcal{G} = (\mathcal{T}, \mathcal{E}, \mathcal{R})$, where $\mathcal{T}, \mathcal{E}, \mathcal{R}$ are the sets of triplets, entities, and relations, respectively. KG language (KGL) is construed as a rigorously defined three-word construct, mirroring the structure of a simple sentence. Specifically, a KGL sentence $e_i r_k e_j$ invariably commences with an entity noun $e_i \in \mathcal{E}$, proceeds with a relation verb $r_k \in \mathcal{R}$, and culminates with another entity noun $e_j \in \mathcal{E}$. Analogous to the syntactic conventions in Chinese, KGL sentences eschew the use of spaces or commas to demarcate KGL terms.

**Knowledge Graph Completion**    KG completion is one of the most important tasks in the KG area. The target of KG completion is to predict the head entity $e_i$ given the relation and tail entity $(?, r_j, e_k)$, or predict the tail entity $e_k$ given $(e_i, r_j, ?)$. In the scenario of KGL, this task is equivalent to completing the KG sentence $?r_j e_k$ or $e_k r_j ?$.

The inductive KG completion focus on completing an unobserved KG $\mathcal{G}_{\text{ind}} = (\mathcal{T}_{\text{ind}}, \mathcal{E}_{\text{ind}}, \mathcal{R}_{\text{ind}})$. Specifically, the relation set $\mathcal{R}_{\text{ind}}$ is identical to the original set $\mathcal{R}$, but the inductive entity set $\mathcal{E}_{\text{ind}}$ shares no elements with $\mathcal{E}$, i.e., $\mathcal{E}_{\text{ind}} \cap \mathcal{E} = \varnothing$. The triplet set $\mathcal{T}_{\text{ind}}$ is further split into the fact set $\mathcal{T}_{\text{ind-fact}}$ and test set $\mathcal{T}_{\text{ind-test}}$. We train a model on the original triplet set $\mathcal{T}$ and use the fact set $\mathcal{T}_{\text{ind-fact}}$ as context to evaluate it on the test set $\mathcal{T}_{\text{ind-test}}$.

**Large Language Models**    As depicted on the left side of Figure 1, the architecture of a typical LLM can be divided into four main components:

- Tokenizer, which breaks down the input sequence of words $w_0, w_1, ..., w_m$ into tokens $t_0, t_1, ..., t_n$;
- Token embedding, which maps the input tokens $t_0, t_1, ..., t_n$ to a sequence of low-dimensional vectors $\mathbf{t}_0, \mathbf{t}_1, ..., \mathbf{t}_n$;
- Transformer $\mathcal{M}$, the core of the LLM, which consists of multiple attention-based blocks that process the input token embeddings into hidden states:

$$\mathbf{h}_0, \mathbf{h}_1, ..., \mathbf{h}_n = \mathcal{M}(\mathbf{t}_0, \mathbf{t}_1, ..., \mathbf{t}_n); \tag{1}$$

- Score layer, which features a weight matrix $\mathbf{S} \in \mathbb{R}^{N \times d}$ with an identical shape to the token embedding matrix $\mathbf{T} \in \mathbb{R}^{N \times d}$, where $N, d$ denote the vocabulary size and hidden size, respectively. The score layer projects the output of Transformer at the $n$-th step to a probability distribution $\mathbf{p}_{n+1}$ for predicting the next token $t_{n+1}$:

$$\mathbf{p}_{n+1} = \mathbf{h}_n \mathbf{S}, \tag{2}$$

## 3.2   Instruct an LLM in KG Language

Recent studies reveal that LLMs harbor the potential to acquire unfamiliar natural languages [46, 47]. Given this premise, it is of particular interest to investigate how LLMs might interpret and operate within our KGL. We first design a prototype instructional text for this purpose. For a given triplet *(Wendee Lee, actor of, Mighty Morphin Power Rangers)*, suppose that the task is to predict the tail entity *Mighty Morphin Power Rangers*, the instructional text is formatted as follows:

**Instruction 3.1.** *Supposed that you are a linguist versed in an esoteric three-word knowledge graph language. Given the following dictionary comprising two words from this language, please kindly*

| Word | Type | Translation | Example |
|------|------|-------------|---------|
| <kgl: Wendee Lee> | entity noun | Wendee Lee | <kgl: Wendee Lee><kgl: profession><kgl: Actor> |
| <kgl: actor of> | relation verb | actor of | <kgl: Peter O'Toole><kgl: actor of><kgl: Gulliver's Travels> |

Table 1: An illustrative KGL-to-English dictionary.

*complete the following sentence: <kgl: Wendee Lee><kgl: actor of>*

Here, <kgl: Wendee Lee> denotes the definitive KGL token (corresponding to $e_i$ in previous sections and Figure 1) assigned to the entity *Wendee Lee*. We enrich the tokenizer's vocabulary with all pertinent KGL tokens, thereby enabling it to translate these KGL tokens into token IDs, which append sequentially to the LLM's original vocabulary range. It is worth noting that we only provide at most one example KGL sentence for each KGL word. Our intention is to introduce the schematics of KGL sentences to the LLM, rather than leveraging augmented KG data for in-context learning. To mitigate potential biases, the example sentences are sampled randomly.

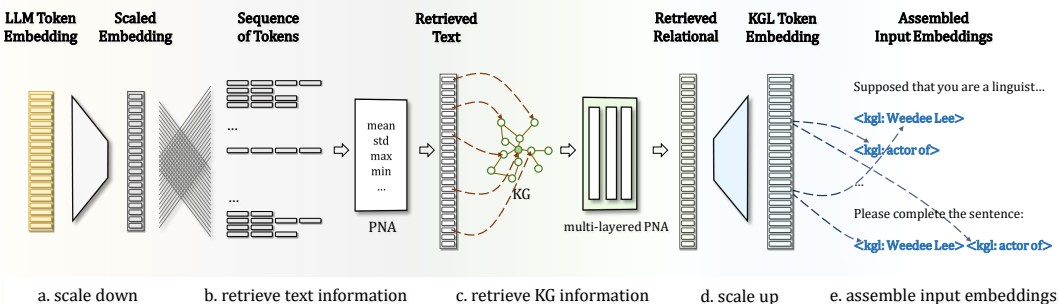

Figure 2: Illustration of LoRA-based KGL Context Retriever. (a) The token embeddings are first scaled down to lower-dimensional vectors; (b) For each input KGL token, their constituting textual token embeddings are aggregated by a PNA encoder; (c) The output embeddings are further aggregated by multi-layered PNA encoders to retrieve neighboring information within KG; (e) The final embeddings are assigned to the KGL tokens.

### 3.3 In-Context Learning versus Special Token Embedding

The practice of incorporating supplementary context information alongside instructional prompts, known as in-context learning (ICL), has proven effective in enhancing performance across many NLP tasks [43, 44]. However, the concatenation of retrieved context on KGs with the input text can easily exceed the input length constraints of LLMs. Processing such long input sequences remains computationally intensive even with truncation. To address these constraints, we propose an alternative approach to encode context information into compact vector representations. Our experiments in Section 4.6 also demonstrate its superiority in terms of both efficiency and performance.

### 3.4 LoRA-based KGL Context Retriever

We propose the low-rank adaption (LoRA)-based KGL context Retriever $R_{\text{context}}$ to effectively aggregate textual and KG information into KGL token embeddings. Typically, the vocabulary scope of a KG (comprising both entities and relations) usually surpasses that of an LLM. For instance, WN18RR is a KG completion dataset set sampled from WordNet [48]. It has over 40,000 unique entities, while the vocabulary size of Llama-2-7b is 32,000. Therefore, initializing new token embeddings for each KG elements and optimizing them from scratch would be prohibitively resource-intensive.

Moreover, the dynamic nature of real-world KGs consistently introduces new entities. This is analogous to the evolution of human language, where new words are often synthesized or derived from existing ones. Drawing inspiration from this linguistic adaptability, we propose leveraging existing textual tokens to generate new KGL tokens, thereby avoiding the computational burden of learning unique embeddings for every KG element.

**Scale Down** As illustrated in Figure 2, the first step is to reduce the dimensionality of LLM token embeddings to lower computational demands during text and KG context aggregation. Inspired by LoRA [12], we leverage a projection matrix $\mathbf{W}_T \in \mathbb{R}^{d \times r}$ to transform the token embedding matrix $\mathbf{T} \in \mathbb{R}^{N \times d}$ into a reduced space $\mathbb{R}^{N \times r}$:

$$\mathbf{T}_r = \mathbf{T}W_T, \tag{3}$$

where $\mathbf{T}_r \in \mathbb{R}^{N \times r}$ represents the compact token embedding matrix.

**Retrieve Text Information** We leverage a text encoder to encode the textual token embeddings of each KGL token into a unified vector. For example, the entity name "Mighty Morphin Power Rangers" would be converted into individual token embeddings $\mathbf{t}_{e_i,0}, \mathbf{t}_{e_i,1}, ..., \mathbf{t}_{e_i,n}$, which are then aggregated into a single vector for the entity $e_i$:

$$\mathbf{t}_{e_i} = \mathcal{E}_{\text{text}}(\mathbf{t}_{e_i,0}, \mathbf{t}_{e_i,1}, ..., \mathbf{t}_{e_i,n}), \tag{4}$$

where $\mathbf{t}_{e_i}$ is the textual token embedding for $e_i$. The choice of the encoder $\mathcal{E}_{\text{text}}$ is free. In this paper, we leverage principal neighbourhood aggregation (PNA) [49], which can be roughly understood as applying multiple pooling operations (including max, min, mean, std etc.) on the token embedding sequences. A detailed introduction to PNA can be found in Appendix C.

**Retrieve KG Information**   We employ a multi-layered PNA encoder $\mathcal{E}_{\text{kg}}$ to aggregate the KG information of $e_i$ and its adjacent entities, which can be formulated as:

$$\mathbf{t}'_{e_i} = \mathcal{E}_{\text{kg}}(\mathbf{t}_{e_i}, \mathcal{N}(e_i)), \tag{5}$$

where $\mathcal{N}(e_i)$ denotes the neighboring entities to $e_i$. The adoption of PNA for encoding both textual and relational data of KGL tokens is due to its parameter efficiency and superior performance compared to attention-based alternatives like GAT [50]. An empirical comparison of different encoders can be found in Appendix F.

**Scale Up**   To finalize, we adjust the dimensionality of the output embeddings to align with the LLM input requirements:

$$\mathbf{t}''_{e_i} = \mathbf{t}'_{e_i} \mathbf{W}_B \tag{6}$$

For the sake of clarity, we will continue to use $\mathbf{t}_{e_i}$ to represent the KGL token embedding in subsequent discussions. For efficiency, we retrieve the KG information only for entities. This operation also make the embeddings of entities and relations distinguishable.

## 3.5   Reconstructing Vocabulary or Constraining the Output Space

While recent studies have adapted LLMs to various tasks by either restricting the output space or reformulating tasks into multiple-choice questions [9, 10, 51–53], such strategies pose challenges for KG completion. Specifically, the existing methods are inapplicable to entity constrastive learning as their main objective is optimized against text tokens instead of entities. Also, they incur significantly slow inference times, as the LLM must traverse to the output tree's leaf nodes to generate predictions. Even then, the generation of top-$k$ results, dependent on beam search parameters, may not accurately reflect the true likelihoods of entities.

In contrast, in this paper we propose a new approach to reconstruct the KGL scores through LLM's score layer and hidden states, providing a one-shot probability distribution for all candidates. Our method seamlessly integrates with contrastive loss and negative sampling techniques [54], making it highly compatible with prevalent KG completion frameworks. This compatibility also ensures that MKGL has the potential of being applied for downstream KG embedding tasks [32, 55].

## 3.6   LoRA-based KGL Score Retriever

We propose a LoRA-based KGL score retriever $R_{\text{score}}$ to produce the probability distribution of KGL tokens, which can be formulated as follows:

$$\mathbf{S}' = \mathbf{S}W_S, \quad \mathbf{h}'_n = \mathbf{h}_n \mathbf{W}_H \qquad \text{(Down Scaling)} \tag{7}$$
$$\mathbf{s}'_j = \mathcal{S}_{\text{text}}(\mathbf{s}'_{j,0}, \mathbf{s}'_{j,1}, ..., \mathbf{s}_{j,n}), \qquad \text{(Text Information Retrieval)} \tag{8}$$
$$\mathbf{s}''_{e_j|e_i,r_k} = \mathcal{S}_{\text{kg}}([\mathbf{h}'_n, \mathbf{s}'_j], \mathcal{N}(e_j)) \qquad \text{(Conditioned Retrieval)} \tag{9}$$
$$p_{e_j|e_i,r_k} = \mathbf{s}''_{e_j|e_i,r_k} \mathbf{W}_O \qquad \text{(Score Estimation)} \tag{10}$$

The score retriever also starts from a down-scaling layer to reduce the dimensionality of the score matrix $\mathbf{S} \in \mathbb{R}^{N \times d}$ to $\mathbf{S}_R \in \mathbb{R}^{N \times r}$ with $\mathbf{W}_S$, and similarly scales down the LLM's output hidden vector $\mathbf{h}_n$ with $\mathbf{W}_H$. Subsequently, the text information (i.e., the token score vectors $\mathbf{s}'_{j,0}, \mathbf{s}'_{j,1}, ..., \mathbf{s}_{j,n}$) associated with target entity $e_j$ is fed to the score text encoder $\mathcal{S}_{\text{text}}$ to construct the KGL score vector $\mathbf{s}'_j$. It is then concatenated with the LLM hidden state $\mathbf{h}'_n$ to obtain the conditioned input $[\mathbf{h}'_n, \mathbf{s}'_j]$. Upon gathering the neighboring information of the target entities via a multi-layered PNA $\mathcal{S}_{\text{kg}}$, an output matrix $\mathbf{W}_O \in \mathbb{R}^{r \times 1}$ is employed to map the result $\mathbf{s}''_{e_j|e_i,r_k} \in \mathbb{R}^r$ to the 1-d probability estimate $p_{e_j|e_i,r_k} \in \mathbb{R}$.

Table 2: The KG completion results on FB15k-237 and WN18RR. The best and second-best results are **boldfaced** and underlined, respectively. ↑: higher is better; ↓: lower is better. -: unavailable entry.

| Model | FB15k-237 | | | | WN18RR | | | |
|---|---|---|---|---|---|---|---|---|
| | MRR↑ | Hits@1↑ | Hits@3↑ | Hits@10↑ | MRR↑ | Hits@1↑ | Hits@3↑ | Hits@10↑ |
| TransE [23] | .310 | .218 | .345 | .495 | .232 | .061 | .366 | .522 |
| RotatE [26] | .338 | .241 | .375 | .533 | .476 | .428 | .492 | .571 |
| TuckER [56] | .358 | .266 | .394 | .544 | .470 | .443 | .526 | .526 |
| CompGCN [28] | .355 | .264 | .390 | .535 | .479 | .443 | .494 | .546 |
| DAN [15] | .354 | .261 | - | .544 | .458 | .422 | - | .537 |
| CoKE [29] | .364 | .272 | .400 | .549 | .484 | .450 | .496 | .553 |
| KG-BERT [14] | - | - | - | .420 | .216 | .041 | .302 | .524 |
| StAR [38] | .296 | .205 | .322 | .482 | .401 | .243 | .491 | .709 |
| KGLM [40] | .289 | .200 | .314 | .468 | .467 | .330 | .538 | .741 |
| FTL-LM [39] | .348 | .253 | .386 | .521 | .543 | .452 | .637 | **.773** |
| DET [30] | .376 | .281 | - | .560 | .507 | .465 | - | .585 |
| KG-Llama-7b [42] | - | - | - | - | - | .242 | - | - |
| GPT 3.5 Turbo [41] | - | .267 | - | - | - | .212 | - | - |
| KICGPT [10] | .412 | **.327** | .448 | .554 | .549 | .474 | **.585** | .641 |
| MKGL | **.415** | .325 | **.454** | **.591** | **.552** | **.500** | .577 | .656 |

**Optimization**   With the above score retriever, estimating the probability for any candidate entity becomes straightforward at a single step. To refine MKGL, we consider a contrastive loss leveraged in most existing KG embedding methods [22, 23, 28, 34], expressed as:

$$\mathcal{L} = \sum_{(e_i, r_k, e_j) \in \mathcal{T}_{\text{train}}} \Big[ -\log(p_{e_j|e_i, r_k}) + \frac{1}{|\mathcal{N}_{\text{neg}}(e_j)|} \sum_{e_{\text{neg}} \in \mathcal{N}_{\text{neg}}(e_j)} \log(1 - p_{e_{\text{neg}}|e_i, r_k}) \Big], \quad (11)$$

where $\mathcal{N}_{\text{neg}}(e_j) = \{e_{\text{neg}}|e_{\text{neg}} \neq e_j, e_{\text{neg}} \in \mathcal{E}\}$ is the sampled negative entity set for the target entity $e_j$. The loss function $\mathcal{L}$ is in a form of a binary cross-entropy, contrasting the likelihood of correctly predicting the relation $p_{e_j|e_i, r_k}$ as positive example, against the probabilities of erroneously predicting relations $(e_i, r_k, e_{\text{neg}})$ as negative examples. We also present an algorithm to demonstrate the step-by-step fine-tuning process, please refer to Appendix D for details.

### 3.7   Complexity

It is clear that the primary computational cost for MKGL lies in the LLM. By employing LoRA-based KGL retrievers to retrieve context vectors instead of texts, we can significantly reduce the major expenditure. For instance, our retrievers can reduce the average input lengths from 811.2 to 91.4 on the FB15k-237 dataset, compared to using one-hop neighbors for in-context learning. All operations within the LoRA-based retrievers are performed under low dimensionality. Furthermore, the token embeddings and score matrix of the LLM are frozen during fine-tuning, thus ignoring their gradient computation. In the worst case, the complexity of text information retrieval is $\mathcal{O}(N_{\text{kgl}}L_{\text{kgl}}r)$, where $N_{\text{kgl}}$, $L_{\text{kgl}}$, $r$ are the number of KGL tokens, maximum text token lengths of KGL tokens, and the reduced dimensionality, respectively. Subsequently, the complexity of KG information retrieval in the worst case is linear to the number of triplets, i.e., $\mathcal{O}(|\mathcal{T}|N_{\text{layer}}r)$, where $|\mathcal{T}|$, $N_{\text{layer}}$ denote the number of triplets in the KG and the number of PNA layer, respectively.

## 4   Experiments

In this section, we evaluate the performance of the proposed MKGL through extensive experiments, comparing it against both LLM-based and KG embedding methods. The source code and datasets are available at github.com/zjukg/MKGL.

### 4.1   Datasets

We evaluate MKGL on the FB15k-237 and WN18RR datasets, which are widely used by most KG completion methods [22, 23, 26, 28, 34, 56, 57]. We also evaluate MKGL on the inductive version of

Table 3: The inductive KG completion results on FB15k-237-ind and WN18RR-ind (v1). The results on other subsets can be found in Appendix F.

| Model | FB15k-237-ind | | | WN18RR-ind | | |
|---|---|---|---|---|---|---|
| | MRR↑ | Hits@1↑ | Hits@10↑ | MRR↑ | Hits@1↑ | Hits@10↑ |
| RuleN [60] | .363 | .309 | .446 | .668 | .635 | .730 |
| NeuralLP [33] | .325 | .243 | .468 | .649 | .592 | .772 |
| DRUM [61] | .333 | .247 | .474 | .666 | .613 | .777 |
| GraIL [58] | .279 | .205 | .429 | .627 | .554 | .760 |
| RED-GNN[59] | .369 | .302 | .483 | .701 | .653 | .799 |
| ChatGPT 3.5 Turbo [41] | - | .288 | - | - | .279 | - |
| MKGL | **.475** | **.400** | **.595** | **.746** | **.700** | **.822** |

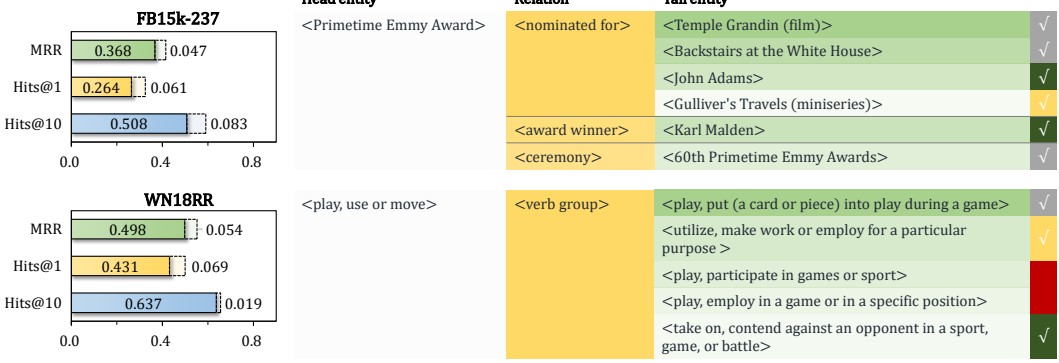

Figure 3: Illustration of KGL modeling. The left shows the performance degradation (in lighter shades) from consecutive predictions of relations and entities. The right presents sentences generated by MKGL, with deeper hues indicating higher probabilities. In the final column, colors grey, green, yellow, and red represent existing, valid, valid but not within the KG, and invalid, respectively.

these two datasets [58]. We follow REDGNN [59] to evaluate MKGL on all entities rather than $50$ sampled candidates. Please refer to Appendix E for dataset statistics.

## 4.2 Settings

For our experiments, we employ Llama-2-7b [11] as the base LLM and train MKGL using $8$ A100 GPUs. A standard LoRA adaptation is applied to the query and value layers of the LLM. Full hyper-parameter details are available in Appendix D. We evaluate performance using MRR (mean reciprocal rank of target entities) and Hits@$k$ (percentage of target entities ranked in the top $k$).

Our baselines include conventional KG embedding methods such as TransE [23], RotatE [26], and TuckER [56]; GNN-based methods like CompGCN [28], DAN [15], and CoKE [29]; methods that integrate language models including KG-BERT [14], StAR [38], KGLM [40], FTL-LM [39], and DET [30]; and LLM-based methods: KG-Llama [42], GPT 3.5 [41], and KICGPT [10]. In the inductive scenario, we compare against rule-based reasoning methods such as RuleN [60], NeuralLP [33], DRUM [61], GraIL [58] and RED-GNN [59], acknowledging that standard methods fail to predict relations without entity embeddings.

## 4.3 Knowledge Graph Completion

The knowledge graph completion results are presented in Table 2. MKGL outperforms other baselines in nearly all metrics. Notably, MKGL and KICGPT significantly surpass other LLM-based methods, demonstrating the importance of KG relational information. Contrarily, many BERT-based methods fall short against GNN-based methods, suggesting that merely incorporating text information may not yield the anticipated benefits. In summary, the proposed MKGL clearly outshines its counterparts, particularly those founded on commercial LLMs.

To our knowledge, existing LLM-based methods have not addressed the inductive KG completion challenge. We benchmark MKGL against the state-of-the-art inductive methods. Although we can

Table 4: Ablation studies on FB15k-237 and WN18RR.

| Score Retriever | | Context Retriever | | FB15K-237 | | | WN18RR | | |
|---|---|---|---|---|---|---|---|---|---|
| Text | KG | Text | KG | MRR↑ | Hits@1↑ | Hits@10↑ | MRR↑ | Hits@1↑ | Hits@10↑ |
| √ | √ | √ | √ | **.415** | **.325** | **.591** | **.552** | **.500** | **.656** |
| | √ | √ | √ | .382 | .294 | .556 | .541 | .482 | .649 |
| | | √ | √ | .365 | .272 | .550 | .512 | .470 | .622 |
| | | | √ | .359 | .260 | .546 | .492 | .437 | .615 |
| | | | √ | .335 | .247 | .535 | .466 | .376 | .574 |

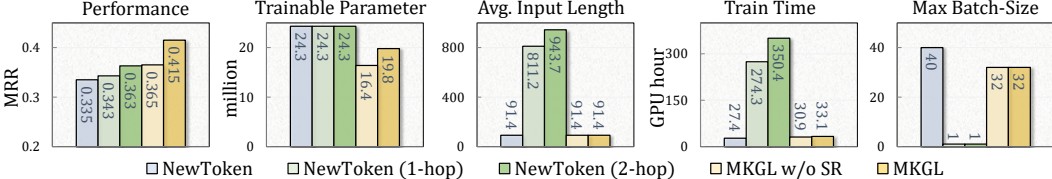

Figure 4: A comprehensive comparison between the methods with randomly-initialized new entity token embeddings (denoted by NewToken) and MKGL on FB15k-237. 1-hop and 2-hop are the versions leveraging the 1-hop and 2-hop KG neighboring information. MKGL w/o SR denotes MKGL without the score retriever.

not assess KICGPT [10] due to unavailable source code, it is worth mentioning that our MKGL could potentially augment KICGPT by supplying a candidate list, facilitating seamless integration between the two methods. We present the results in Table 3. MKGL significantly outperforms all the baseline methods across metrics. As most inductive reasoning methods do not have an embedding module for entities, the proposed MKGL represents the first embedding-based method to deliver state-of-the-art results in inductive KG completion.

## 4.4 Knowledge Graph Language Modeling

Beyond its capability as a KG completion tool, MKGL also serves as a language model for KG languages. To evaluate its proficiency in generating KGL sentences, we employ a sequence generation loss and remove the relation context from the input prompts. We leverage the second-to-last output of the LLM for relation prediction.

The results are shown in Figure 3. The left section contrasts the sequence prediction results against standard KG completion, revealing only a modest loss in performance. MKGL still outperforms many conventional methods, especially on WN18RR dataset. The right panel displays sample sentences generated by MKGL, illustrating its potential to discover legitimate KGL sentences absent from the existing KG. We observe that WN18RR is more difficult than anticipated as it contains many plausible entities that challenge even an LLM's discernment.

## 4.5 Ablation Study

We conduct ablation studies to assess the importance of each module, as detailed in Table 4. The unmarked cells indicate that we either substitute the text retrieval module with a learnable embedding module or remove the KG retrieval module. Clearly, the method with complete features achieves best results, while replacing or removing either module significantly impacts performance. Notably, removing the KG retrieval module yields more performance loss on WN18RR, as many entities in this dataset have similar names. For example, there are 14 different entities named "call". In this case, incorporating with KG information becomes necessary.

## 4.6 Computational Cost

We examine the computational efficiency of our method (MKGL) relative to "in-context" baselines. Specifically, we develop several variants: LLM randomly-initialized new entity token embeddings (NewToken), LLM with KGL context from 1-hop neighbors (NewToken (1-hop)), LLM with KGL context from 2-hop neighbors (NewToken (2-hop)), and MKGL without score retriever (MKGL w/o

SR). The results are shown in Figure 4. MKGL surpasses all alternatives in performance. NewToken variants slightly lag behind MKGL w/o SR, but notably, our proposed methods demand fewer trainable parameters than NewToken variants. By encoding all context information within KGL token embeddings, the average input length is significantly reduced, which decreases training time considerably. Moreover, MKGL supports larger batch sizes during both training and inference phases, enhancing computational efficiency.

## 5    Conclusion and Future Work

In this paper, we propose MKGL to instruct the LLM in the language of KGs. MKGL employs a context retriever that efficiently provides LLMs with pertinent textual and relational context, markedly reducing input lengths relative to in-context-learning and supervised fine-tuning methods. Meanwhile, MKGL also leverages a score retriever to supply score information and aid in KGL inference. Extensive experiments confirm the superiority of MKGL in terms of both performance and computational efficiency. The proposed context and score retrievers point out a new direction in incorporating LLMs with semantic data, such as question answering and entity linking. They may also shed lights on a more broaden area where the input cannot be precisely represented by text, e.g., node classification and protein representation learning. Furthermore, the construction of KGL vocabulary enables contrastive learning not only limited on tokens, which may provide insights on general machine learning. Therefore, there are plenty of future directions. We would like to pretrain LLM using the mixture corpora of KG and natural languages, such that the LLM could understand and create responses with linked data.

## Acknowledgement

This work is founded by National Natural Science Foundation of China (NSFC62306276/NSFCU23B2055/NSFCU19B2027/NSFC6240072039), Zhejiang Provincial Natural Science Foundation of China (No. LQ23F020017), Yongjiang Talent Introduction Programme (2022A-238-G), and Fundamental Research Funds for the Central Universities (226-2023-00138). This work was supported by AntGroup.

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

# A  Limitations

We would like to discuss the potential limitations of our method from the following three aspects:

Efficiency. As MKGL is an LLM-based fine-tuning method, it inevitably demands more computational resources. In the main experiments, MKGL significantly outperforms all the conventional and LLM-based methods. The later analysis also reveal that the trainable parameters and runtime of MKGL are less than general fine-tuning framework. Therefore, we believe that MKGL is still an efficient LLM-based method.

Robustness. MKGL leverage multiple retrievers to retrieve text and KG information for constructing both input embeddings and score estimations, which may accumulate more errors during fine-tuning. Even though, most modules are learnable with back-propagation. To avoid biased evaluation and occasional results, we also report the averaged results of multiple runs with variance statistics. Thus, we believe MKGL is a robust method.

Generality. The advances in LLMs have revolutionized many NLP tasks, and it is important for an LLM-based method where the LLM makes use of the proposed modules and whether the performance can continually improves as the LLM get promotion. We have conducted experiments to visualize the KGL embeddings and compare the performance with different base LLMs. The results empirically demonstrate the generality and potential of MKGL.

# B  Broader Impacts

Our work focuses on the integration of Large Language Models (LLMs) with Knowledge Graphs (KGs) through the introduction of a specialized KG Language (KGL), has substantial broader impacts spanning technological advancements, societal implications, and educational benefits. Here, we outline the diverse and far-reaching impacts of our research.

**Technological Advancements**  Our research contributes to the cutting-edge of artificial intelligence, pushing the boundaries of what LLMs can achieve when combined with the structured knowledge represented by KGs. This can potentially unlock new capabilities in AI, ranging from more accurate context-aware natural language understanding to enhanced machine reasoning across diverse domains such as healthcare, finance, and legal systems. Additionally, our approach of using a specialized language (KGL) and the method of contextual embedding augmentation can inspire novel AI architectures and learning paradigms that bridge the gap between unstructured and structured forms of knowledge.

**Societal Implications**  The enhancement of AI systems with a more profound understanding of structured knowledge has broad societal implications. For one, it could lead to the development of AI assistants that provide more accurate, consistent, and reliable information, thus improving decision-making in critical sectors. In healthcare, for instance, AI systems equipped with our technology could offer more precise recommendations by thoroughly understanding medical knowledge graphs. Moreover, by reducing the propensity for errors and hallucinations, our approach could foster greater trust in AI technologies among the general public, paving the way for wider acceptance and integration into daily life.

**Ethical Considerations**  As with any advancement in AI, our work prompts important ethical considerations. Ensuring our technology is used responsibly involves critical discussions around privacy, bias, and transparency, especially as AI systems become more adept at interpreting and generating human-like text. We advocate for the continued examination of these aspects in tandem with technological development to ensure AI benefits society equitably and ethically. Our work, by facilitating error reduction in AI outputs, also contributes to the broader effort of minimizing harm and bias in AI-generated content.

**Educational Benefits**  Our integration of LLMs with KGs presents a novel avenue for educational tools and applications. AI tutors or educational platforms powered by our enhanced LLMs can offer students personalized and accurate learning experiences. Such systems could better understand and integrate complex academic content structured within knowledge graphs, from history timelines to

scientific concepts, thereby improving the quality of automated tutoring services and opening up new methods for engaging with educational material.

**Future Directions**   This research opens up exciting avenues for future exploration, including refining the KGL for broader applicative scopes, exploring the ethical considerations of nuanced AI-generated content, and expanding our understanding of AI's potential when it deeply integrates diverse forms of knowledge. The cross-disciplinary nature of this work invites collaboration among computer scientists, ethicists, domain experts, and policymakers to harness the full potential of AI for societal benefit.

In conclusion, the integration of LLMs with KGs not only represents a significant step forward in AI capabilities but also poses thoughtful considerations for societal impact, ethical use, and educational applications. Our work underscores the importance of continuous exploration, responsible innovation, and cross-disciplinary collaboration to harness the transformative potential of AI technologies.

## C   Principal Neighborhood Aggregation

Graph Neural Networks (GNNs) have emerged as a powerful family of neural models for learning on graph-structured data [17, 18, 62]. Among the recent advances is the principal neighborhood aggregation (PNA) mechanism [49], which enhances the representational capacity of GNNs by diversifying the aggregation functions applied to neighboring nodes.

PNA leverages a combination of multiple aggregators such as sum, mean, and standard deviation, together with a scalable degree-specific weighting scheme. This approach is designed to address the shortcomings associated with simple aggregation functions that may fail to capture the complexity and diversity of neighborhood structures in graphs.

The key component of PNA is its aggregation scheme, which is formally defined as follows:

$$a_v^{(l+1)} = \delta \left( \bigoplus_{\rho \in \mathcal{R}} AGG_\rho(\{h_u^{(l)}, \forall u \in \mathcal{N}(v)\}), W^{(l)} \right) \quad (12)$$

Please note that the symbols used in describing PNA are independent to the main paper for clarity. Here, $a_v^{(l+1)}$ is the aggregated information for node $v$ at layer $l+1$, $\mathcal{N}(v)$ denotes the set of neighbors of $v$, $h_u^{(l)}$ represents the hidden features of neighbor nodes at layer $l$, $\bigoplus$ is a concatenation operator over all aggregators in the set $\mathcal{R}$, $AGG_\rho$ is an aggregation function (e.g., sum, mean, max), $\delta$ is a nonlinear activation function such as ReLU, and $W^{(l)}$ is a learnable weight matrix at layer $l$.

$$a_v^{(l+1)} = \delta \left( \bigoplus_{\rho \in \mathcal{R}} AGG_\rho(\{h_u^{(l)} \otimes h_r^{(l)}, \forall (v, r, u) \in \mathcal{T}\}), W^{(l)} \right) \quad (13)$$

PNA's distinctive blend of multiple aggregation functions and degree-aware weighting significantly enhances the expressive power of GNNs, allowing for more complex feature representations and, consequently, improved performance on downstream tasks. We also follow the KG embedding methods [15, 28, 63] to incorporate the relation embeddings into PNA as relational PNA.

## D   Implementation Details

We introduce Algorithm 1 to demonstrate the fine-tuning process of MKGL for KG completion. We first construct input instructions following Instruction 3.1 and tokenize them into IDs. For those in the score of original LLM vocabulary, their embeddings can be looked up from **T**, while those of out of scope will be retrieved by our context retriever $R_{\textbf{context}}$. After assembling the input embeddings, we feed them to the LLM to obtain output hidden states and then obtain the scores from the score retriever $R_{\text{score}}$. Finally, we optimize MKGL by minimizing the constrastive loss $\mathcal{L}$. The main hyper-parameter settings are summarized in Table 5.

**Algorithm 1** MKGL for KG Completion
---
1: **Input:** the training KG $\mathcal{G}$, the language model $\mathcal{M}$, the token embedding matrix $\mathbf{T}$, the original vocabulary of the LLM $\mathcal{V}_{\text{llm}}$, the KGL context retriever $R_{\text{context}}$, the KGL score retriever $R_{\text{score}}$;
2: **for each** batched data in the training set **do**
3:     Construct input instructions according to Instruction 3.1;
4:     Tokenize the input instructions with the tokenizer;
5:     **for each** input token sequence $\{t_0, t_1, t_2, ...\}$ **do**
6:         **for each** token $t_k$ **do**
7:             **if** $t_k \in \mathcal{V}_{\text{llm}}$ **then**
8:                 $\mathbf{t}_k \leftarrow \mathbf{T}_k$;    *// look up embedding from the token matrix*
9:             **else**
10:                $\mathbf{t}_k \leftarrow R_{\text{context}}(t_k)$ (Equations (3-6));
11:             **end if**
12:         **end for**
13:     **end for**
14:     Compute the batched output hidden states of the LLM $\mathcal{M}$ (Equation 1);
15:     Compute the batched scores with $R_{\text{score}}$ (Equations (7-10));
16:     Compute and minimize the constrastive loss $\mathcal{L}$ (Equation (11));
17: **end for**

Table 5: Hyper-parameter settings in the main experiments.

| Datasets | LLM | LoRA r | LoRA dropout | LoRA target modules | train batch size per device | loss criterion | gradient accumulation steps | optimizer |
|---|---|---|---|---|---|---|---|---|
| FB15k-237 | Llama-2-7b-chat | 32 | 0.05 | query, value | 32 | BCE | 1 | Adam 8bit |
| WN18RR | Llama-2-7b-chat | 32 | 0.05 | query, value | 16 | BCE | 4 | Adam 8bit |

| | # epoch | # context retriever r | # context text encoder layer | # context KG encoder layer | # score retriever r | # score text encoder layer | # score KG encoder layer | learning rate (Lora/MKGL) |
|---|---|---|---|---|---|---|---|---|
| FB15k-237 | 5 | 32 | 1 | 6 | 32 | 1 | 6 | 0.0005/0.005 |
| WN18RR | 2 | 32 | 1 | 6 | 32 | 1 | 6 | 0.0001/0.001 |

# E  Dataset Details

We use the following benchmark datasets to evaluate the performance of MKGL, and summarize the statistics in Table 6:

- **FB15k-237:** This dataset is a subset of the original FB15k dataset [23] and is created by removing inverse relations that may lead to test set leakage.

- **WN18RR:** This dataset is a subset of the original WN18 dataset [23] and is created by removing inverse relations that may lead to test set leakage.

- **FB15k-237-ind:** The 'ind' suffix denotes the inductive setting adopted in FB15k-237 [58]. It includes new entities in the validation and test sets that are not present during training, thus requiring models to generalize beyond the transductive assumptions of previously seen entities.

- **WN18RR-ind:** Similarly to FB15k-237-ind, the WN18RR-ind dataset is adapted for inductive KG completion on the WordNet [48].

These datasets have been instrumental in the development and benchmarking of advanced KG completion models, enabling comparison of different approaches and understanding of their effectiveness in both conventional and inductive settings.

Table 6: Dataset statistics.

| Dataset | # Relation | Train | | Valid | | | Test | | |
|---|---|---|---|---|---|---|---|---|---|
| | | # Entity | # Triplet | # Entity | # Evaluation | # Fact | # Entity | # Evaluation | # Fact |
| FB15k-237 | 237 | 14,541 | 272,115 | - | 17,535 | - | - | 20,466 | - |
| WN18RR | 11 | 40,943 | 86,835 | - | 3,034 | - | - | 3,134 | - |
| FB15k-237-ind-v1 | 180 | 1,594 | 4,245 | 1,594 | 489 | 4,245 | 1,093 | 205 | 1,993 |
| FB15k-237-ind-v2 | 200 | 2,608 | 9,739 | 2,608 | 1,166 | 9,739 | 1,660 | 478 | 4,145 |
| FB15k-237-ind-v3 | 215 | 3,668 | 17,986 | 3,668 | 2,194 | 17,986 | 2,501 | 865 | 7,406 |
| FB15k-237-ind-v4 | 219 | 4,707 | 27,203 | 4,707 | 3,352 | 27,203 | 3,051 | 1,424 | 11,714 |
| WN18RR-ind-v1 | 9 | 2,746 | 5,410 | 2,746 | 630 | 5,410 | 922 | 188 | 1,618 |
| WN18RR-ind-v2 | 10 | 6,954 | 15,262 | 6,954 | 1,838 | 15,262 | 2,757 | 441 | 4,011 |
| WN18RR-ind-v3 | 11 | 12,078 | 25,901 | 12,078 | 3,097 | 25,901 | 5,084 | 605 | 6,327 |
| WN18RR-ind-v4 | 9 | 3,861 | 7,940 | 3,861 | 934 | 7,940 | 7,084 | 1,429 | 12,334 |

Table 7: The detailed inductive KG completion results, where v1-v4 represent four different subsets.

| Method | FB15k-237-ind-v1 | | | FB15k-237-ind-v2 | | |
|---|---|---|---|---|---|---|
| | MRR↑ | Hits@1↑ | Hits@10↑ | MRR↑ | Hits@1↑ | Hits@10↑ |
| GraIL [58] | .279 | .205 | .429 | .276 | .202 | .424 |
| NeuralLP [33] | .325 | .243 | .468 | .389 | .286 | .586 |
| DRUM [61] | .333 | .247 | .474 | .395 | .284 | .595 |
| RED-GNN[59] | .369 | .302 | .483 | .469 | .381 | .629 |
| MKGL | **.475** | **.400** | **.595** | **.508** | **.417** | **.681** |

| | FB15k-237-ind-v3 | | | FB15k-237-ind-v4 | | |
|---|---|---|---|---|---|---|
| | MRR↑ | Hits@1↑ | Hits@10↑ | MRR↑ | Hits@1↑ | Hits@10↑ |
| GraIL [58] | .251 | .165 | .424 | .227 | .143 | .389 |
| NeuralLP [33] | .400 | .309 | .571 | .396 | .289 | .593 |
| DRUM [61] | .402 | .308 | .571 | .410 | .309 | .593 |
| RED-GNN[59] | .445 | .351 | .603 | .442 | .340 | .621 |
| MKGL | **.486** | **.392** | **.643** | **.471** | **.374** | **.645** |

| | WN18RR-ind-v1 | | | WN18RR-ind-v2 | | |
|---|---|---|---|---|---|---|
| | MRR↑ | Hits@1↑ | Hits@10↑ | MRR↑ | Hits@1↑ | Hits@10↑ |
| GraIL [58] | .627 | .554 | .760 | .625 | .542 | .776 |
| NeuralLP [33] | .649 | .592 | .772 | .635 | .575 | .749 |
| DRUM [61] | .666 | .613 | .777 | .646 | .595 | .747 |
| RED-GNN[59] | .701 | .653 | .799 | .690 | .633 | .780 |
| MKGL | **.746** | **.700** | **.822** | **.712** | **.662** | **.799** |

| | WN18RR-ind-v3 | | | WN18RR-ind-v4 | | |
|---|---|---|---|---|---|---|
| | MRR↑ | Hits@1↑ | Hits@10↑ | MRR↑ | Hits@1↑ | Hits@10↑ |
| GraIL [58] | .323 | .278 | .409 | .553 | .443 | .687 |
| NeuralLP [33] | .361 | .304 | .476 | .628 | .583 | .706 |
| DRUM [61] | .380 | .330 | .477 | .627 | .586 | .702 |
| RED-GNN[59] | .427 | .368 | .524 | .651 | .606 | .721 |
| MKGL | **.456** | **.406** | **.559** | **.664** | **.620** | **.741** |

# F    Additional Experiment Results

## F.1    More Examples on Knowledge Graph Language Modeling

We present additional examples of KGL modeling in Table 5, which demonstrates that MKGL can not only generate KGL sentences seen during training but also produce previously unseen triplets within the testing set.

## F.2    Details Results on Inductive Knowledge Graph Completion

We present detailed results on all inductive KG completion benchmarks in Table 7, where MKGL consistently and significantly outperforms all state-of-the-art baselines.

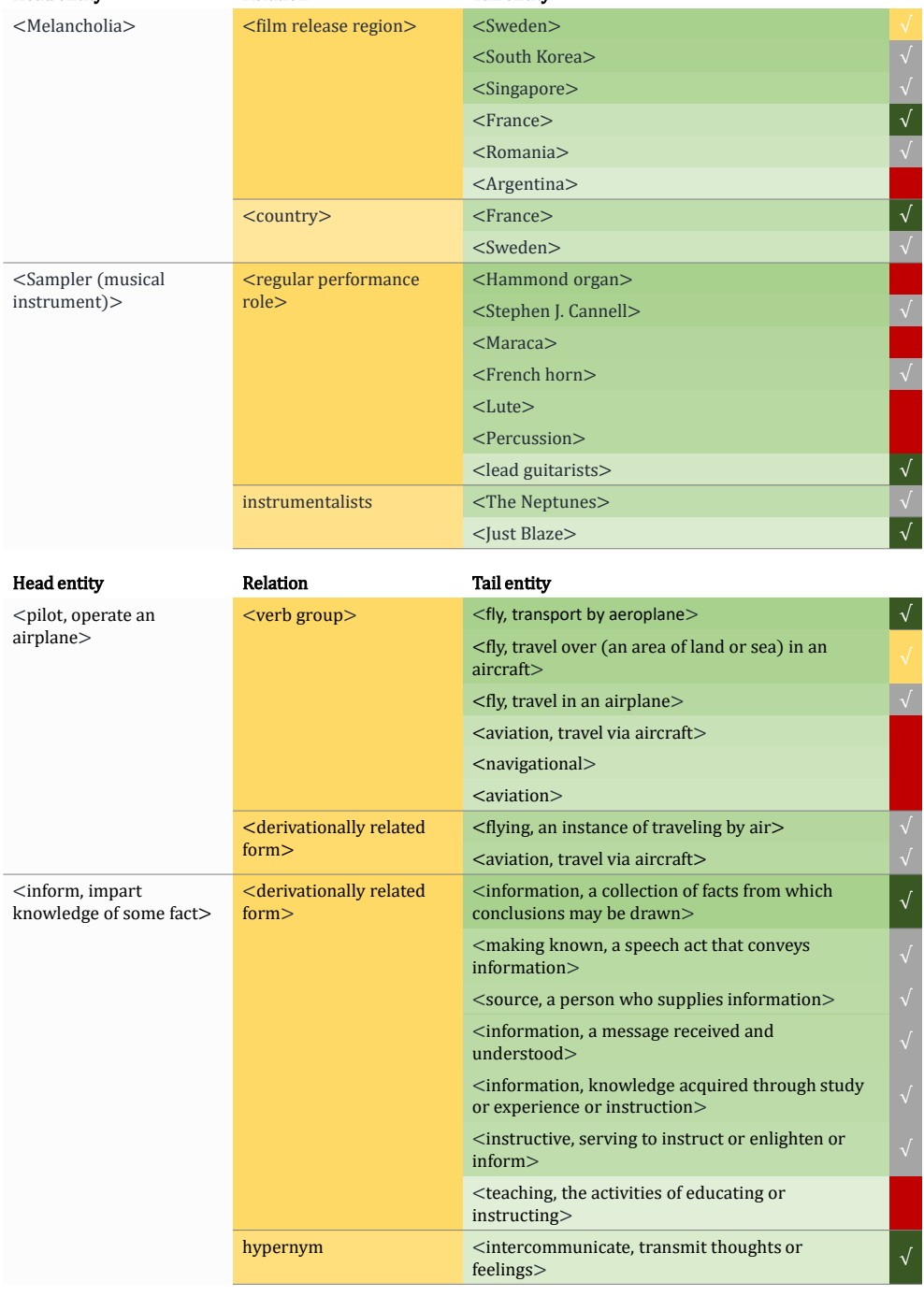

Figure 5: More examples on KGL modeling.

## F.3 Different Layer Numbers

We conduct experiments to analyze the influence of layer numbers in the KGL retrievers. The results are illustrated in Figure 6. Clearly, increasing the number of layers enhances performance across all datasets and metrics. Additionally, we observe that a small number of layers (i.e., 2) significantly impairs performance.

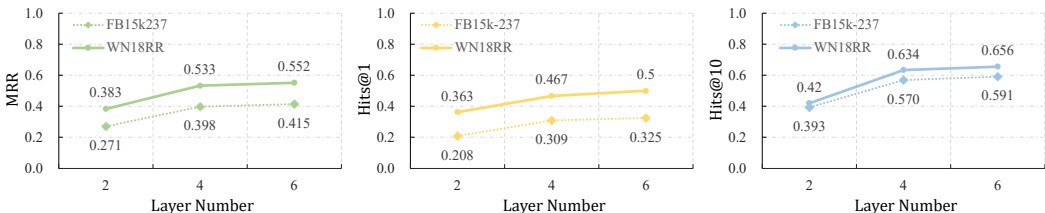

Figure 6: The performance of MKGL on FB15k-237 and WN18RR, with respect to the layer number of the retrievers.

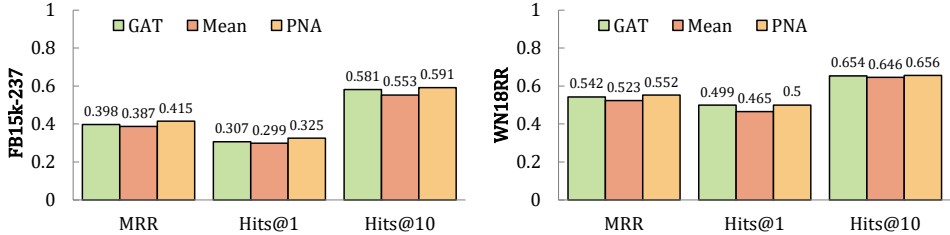

Figure 7: The performance of MKGL on FB15k-237 and WN18RR, with respect to different encoders in the retrievers.

Table 8: Detailed results of Table 2. The KG completion results on FB15k-237 and WN18RR. The best and second-best results are **boldfaced** and underlined, respectively. ↑: higher is better; ↓: lower is better. -: unavailable entry.

| Model | FB15k-237 | | | | | WN18RR | | | | |
|---|---|---|---|---|---|---|---|---|---|---|
| | # Tr. Params | MRR↑ | Hits@1↑ | Hits@3↑ | Hits@10↑ | # Tr. Params | MRR↑ | Hits@1↑ | Hits@3↑ | Hits@10↑ |
| TransE [23] | 2M | .310 | .218 | .345 | .495 | 21M | .232 | .061 | .366 | .522 |
| RotatE [26] | 15M | .338 | .241 | .375 | .533 | 20M | .476 | .428 | .492 | .571 |
| TuckER [56] | 11M | .358 | .266 | .394 | .544 | 9M | .470 | .443 | .526 | .526 |
| CompGCN [28] | 10M | .355 | .264 | .390 | .535 | 12M | .479 | .443 | .494 | .546 |
| DAN [15] | - | .354 | .261 | - | .544 | - | .458 | .422 | - | .537 |
| CoKE [29] | 10M | .364 | .272 | .400 | .549 | 17M | .484 | .450 | .496 | .553 |
| KG-BERT [14] | 110M | - | - | - | .420 | 110M | .216 | .041 | .302 | .524 |
| StAR [38] | 355M | .296 | .205 | .322 | .482 | 355M | .401 | .243 | .491 | .709 |
| KGLM [40] | 355M | .289 | .200 | .314 | .468 | 355M | .467 | .330 | .538 | .741 |
| FTL-LM [39] | 125M | .348 | .253 | .386 | .521 | 125M | .543 | .452 | .637 | **.773** |
| DET [30] | 16M | .376 | .281 | - | .560 | 24M | .507 | .465 | - | .585 |
| KG-Llama-7b [42] | - | - | - | - | - | 13M | - | .242 | - | - |
| GPT 3.5 Turbo [41] | - | - | .267 | - | - | - | - | .212 | - | - |
| KICGPT [10] | - | .412 | **.327** | .448 | .554 | - | .549 | .474 | **.585** | .641 |
| MKGL | 20M | **.415**±.002 | .325±.004 | **.454**±.001 | **.591**±.001 | 20M | **.552**±.002 | **.500**±.005 | .577±.003 | .656±.002 |

## F.4 Different Encoders

We conduct experiments to explore the impact of different encoders in the retrievers. The results are depicted in Figure 7. We find that the MKGL is not highly sensitive to the choice of encoders. The performance when using GAT [50] is slightly lower than when using PNA.

