# OpenReview forum: "MKGL: Mastery of a Three-Word Language"
_NeurIPS.cc/2024/Conference — NeurIPS 2024 spotlight_

### Official Review · Reviewer_NCjt · 2024-07-05

**Soundness:** 2
**Presentation:** 3
**Contribution:** 3
**Rating:** 7
**Confidence:** 3

**Summary:**

This paper proposes a method to leverage LLMs as knowledge graph completion systems. New tokens that correspond to (potentially multi-word) concepts and relations are introduced to the model’s vocabulary, and then the LLM’s embeddings for the tokens composing those concepts/relations are aggregated and upscaled to estimate embeddings for these new tokens. Given these new tokens, the goal of the system is to complete the knowledge graph triplet given two of these embeddings by retrieving the correct third token from the KG vocabulary.

In a series of experiments, it is observed that the proposed system outperforms a variety of previous baselines employing diverse methods. It also excels at inductive KG completion. An ablation study reveals that each proposed part of the pipeline is necessary for achieving the best performance.

**Strengths:**

* The proposed approach outperforms a wide variety of prior methods with respect to accuracy. It also achieves a better trade-off between accuracy and compute-efficiency.
* A wide variety of baselines employing diverse methods are compared.
* Informative ablation study.

**Weaknesses:**

1. Given how many moving parts there are, reproducibility seems difficult. It would be nice to see the variance in performance of the method across multiple random seeds, where each random seed entails running the entire pipeline of optimizations from scratch.
2. Relatedly, it is unclear whether the approach will scale as LMs continue to improve (and presumably to become better bases for approaches like this). Having a comparison with other base models would be a nice way to hedge against this.
3. No detailed discussion of limitations. The checklist says it is discussed, but there are only brief comments distributed throughout the paper (which, in my opinion, do not address limitations thoroughly enough). A dedicated section would be helpful.
4. It is unclear whether the new KG embeddings encode relevant concepts to the target token, or whether they are picking up on certain spurious correlations that happen to be helpful (but may not generalize robustly). It would be nice to have an analysis where the new embeddings are directly decoded into vocabulary space, such that we can observe what concepts are included in these new representations.

**Questions:**

1. Would it be possible to base this approach on other LMs as well? The various Llama scales would be ideal, but if scale is an issue, even just comparing Mistral, Llama 2 (7B), Llama 3 (8B), and ideally some smaller models would be nice. This is quite compute-intensive, so I wouldn’t expect it for the camera-ready, but it would definitely be nice to have.
2. Do you have any hypotheses as to why the proposed method is better at inductive KG completions than prior methods? In other words, is there a particular aspect of your pipeline that you believe makes it better for handling novel triplets than past approaches?

Typos:
* L226: “clear that The” -> “clear that the”

**Limitations:**

I do not believe limitations have been sufficiently addressed. There is no "Limitations" section, nor is there a dedicated space in any part of the paper that directly addresses the drawbacks of the proposed method and experiments. For example, there are many moving parts; there are multiple stages of optimization that could lead to cascading errors; only one LLM base was considered; etc.

---

> ### Author Rebuttal · Authors · 2024-08-06
>
> Thank you very much for your detailed and insightful comments. We have addressed your concerns below and hope our responses provide clarity:
>
> ### Weaknesses:
>
> 1. **Reproducibility: can the authors provide the variance in performance of the method?**
>
>     Thanks for your suggestion. We follow the existing methods to report the average results of 5 runs in the main tables. Most KG completion methods are not very sensitive to the initialization seed [1], as the large label space and testing sets make the result quite stable. This may be the reason why they did not provide the variance statistics. Re-producing all baseline results is difficult at this phase, but we are willing to update the tables with the variance statistics of our method:
>
>     | Methods  | FB15k-237 MRR | FB15k-237 Hits@1 | FB15k-237 Hits@3 | FB15k-237 Hits@10 | WN18RR MRR | WN18RR Hits@1 | WN18RR Hits@3 | WN18RR Hits@10 |
>     |---|:---:|:---:|:---:|:---:|:---:|:---:|:---:|:---:|
>     | MKGL  | .415 $\pm$ .002 | .325 $\pm$ .004 | .454 $\pm$ .001 | .591 $\pm$ .001 | .552 $\pm$ .002 | .500 $\pm$ .005  | .577 $\pm$ .003  | .656 $\pm$ .002 |
>
>
> 2. **How does MKGL perform as LMs continue to improve?**
>
>     Thanks very much for your suggestion. We have conducted a new experirment investigating the impact of the base model, where we consider Mistral (7B), Llama 2 (7B), Llama 3 (8B), and LLama 2 (13B) for comparison. The results are shown the following table:
>
>     | Methods  | FB15k-237 MRR | FB15k-237 Hits@1 | FB15k-237 Hits@3 | FB15k-237 Hits@10 |
>     |---|:---:|:---:|:---:|:---:|
>     | Llama-2-7B  | .415 | .325 | .454 | .591 |
>     | Llama-2-13B  | .421 | .331 | .455 | .590 |
>     | Llama-3-8B  | .429 | .337 | .459 | .593 |
>     | Mistral-7B |  .424 | .334 |  .457 | .591 |
>
>     From the results, we can find the improvement from 7B model to 13B model is limited, possibly because 13B is not significantly larger than 7B. However, more advanced base models do contribute to better performance, which can be empirically verified from the results of Llama-3-8B and Mistral-7B. The results are significantly better than the Llama-2-7B version, especially on Hits@1. They also outperform the best baseline method KICGPT (.327) in Table 2. We have added a subsection to include the above results and discussion in the revision.
>
> 3. **A dedicated limitation section would be helpful.**
>
>     Thank you for the detailed comments. We have incorporated a dedicated limitations section in the revised version, as outlined below:
>
>     We would like to discuss the potential limitations of our method from the following three aspects:
>
>     Efficiency: As MKGL is an LLM-based fine-tuning method, it inherently requires more computational resources. In our primary experiments, MKGL significantly outperformed all conventional and LLM-based methods. Subsequent analyses also revealed that the trainable parameters and runtime of MKGL are fewer than those of a general fine-tuning framework. Hence, we believe that MKGL remains an efficient LLM-based method.
>
>     Robustness: MKGL leverages multiple retrievers to gather text and KG information for constructing both input embeddings and score estimations, which may introduce more errors during fine-tuning. Nevertheless, most modules are learnable through back-propagation. To mitigate biased evaluation and sporadic results, we also present the averaged results of multiple runs alongside variance statistics. Consequently, we consider MKGL to be a robust method.
>
>     Generality: The advancements in LLMs have revolutionized many NLP tasks, and it is important for an LLM-based method whether the proposed module is effective and the performance can continually improve as the LLM gets promoted. We have conducted experiments to visualize the KGL embeddings and compare the performance with different base LLMs. The results empirically demonstrate the generality and potential of MKGL.
>
>
>
> 4. **It would be nice to have an analysis about the new KGL embeddings and original token embeddings.**
>
>     We have visualized the embeddings of the original LLM tokens and KGL tokens in Figure 2 of the newly uploaded rebuttal.pdf. In Figure 2a, we observe that two types of embeddings are (nearly) uniformly distributed in the space. The KGL tokens have been successfully encoded into the original token space. In Figure 2b, we present a sample from the center, where the entity *Canal+* (a sports TV channel) is closely encoded to tokens like *TV* and *_Team*. In Figure 2c, we present a sample from the corner, which also demonstrates high semantic correlations.
>
>     We have included the figure and discussion in the revision. Additionally, we believe that the inductive KG completion experiments could further demonstrate the robustness and generality of MKGL, as the entities in the testing set are unseen and unknown during fine-tuning.
>
> ### Questions:
>
> 1. **Would it be possible to base this approach on other LMs as well? Even just comparing Mistral, Llama 2 (7B), Llama 3 (8B) would be nice.**
>
>     Please refer to our response to the weaknesses.
>
>
> 2. **Why is MKGL better at inductive KG completions than prior methods?**
>
>     We believe there are two possible reasons: Firstly, the incorporation of additional text information. Previous works primarily rely on structural similarities for inductive KG completion, while our method additionally leverages text information. Although the entities in the testing set are new, their text token features are not new to MKGL. Secondly, the utilization of LLM. Without loss of generality, we believe that LLM possesses knowledge helpful in KG completion and offers better inference ability compared to smaller models.
>
>
> - **Typos: L226, “clear that The”**
>
>     Many thanks. We have fixed it.
>
> - **Limitations**
>
>     Thank you once again for the detailed and constructive comments. We hope our response has sufficiently addressed these limitations.
>
> [1]  A Re-evaluation of Knowledge Graph Completion Methods. ACL, 2020.

---

> > ### Comment · Reviewer_NCjt · 2024-08-08
> >
> > Thanks for the thorough response. These low variances are nice to see, and help me trust the robustness of the results.  Thanks also for running the experiments on various Llama versions and sizes.
> >
> > The new embeddings visualization is nice to see, but it presents only a couple examples, and does not directly compare to the original representations. I feel that a more systematic quantitative comparison would better address Weakness 4; this could, for example, be based on the average distance between entity or relation tokens that co-occur in queries in the original vs. new space.
> >
> > That said, I consider the other weaknesses to be well-addressed, even if preliminarily. I'm therefore raising my score.

---

> > > ### Author Response · Authors · 2024-08-09
> > >
> > > We sincerely appreciate your increased rating and recognition of the efforts we put into addressing your concerns. A more straightforward comparison between two types of embeddings is indeed helpful. We believe  the Wasserstein metric (also known as Earth mover’s distance, the cost of an optimal transport plan to convert one distribution into another) is appropriate for estimating the similarity between two distributions. We sampled 1,000 tokens/entities to estimate the Wasserstein distance and present the results in the following table:
> > >
> > > | (X, Y) | Wasserstein distance |
> > > |---|:---:|
> > > | (Token, Token)  | 1.4418 |
> > > | (Token, Entity)  | 2.0019 |
> > >
> > > The Wasserstein distance between the sampled entity and token distributions is slightly larger than that of the two sampled token distributions. Therefore, it is reasonable to conclude that our method has successfully encoded the new KGL tokens into the original token embedding space. We will update the corresponding paragraph to discuss the Wasserstein distance results. Thank you again for your prompt and kind response.

---

### Official Review · Reviewer_dtBb · 2024-07-06

**Soundness:** 2
**Presentation:** 2
**Contribution:** 3
**Rating:** 6
**Confidence:** 2

**Summary:**

This paper proposes what seems to be an elaborate GNN+LLM+GNN  sandwich of a model for doing knowledge base completion, having a GNN pipeline to form KB-informed token embeddings, passing those to a LLM (lllama-2 ) into a knowledge base completion prompt template, and then passing that output into another GNN-like ("PNA") set of layers.  That whole set-up is then used to train representations of the data optimized with contrastive loss for the entity prediction task in knowledge base completion tasks (FBK and Wordnet)   (I'll admit that I found the model explanation quite hard to follow, and so it's entirely possible I'm slipping on some details -- some of this had to be inferred by glancing at their code.)  It seems to outperform existing methods on these two tasks.

**Strengths:**

If the authors work and evaluation are sound, their method outperforms other methods at two commonly used knowledge base completion tasks.

**Weaknesses:**

- I found the model extremely rather hard to follow, and had particular trouble discerning why this collection of model assumptions would result in a meaningful improvement over prior work.  Since the work is so complicated, it may be both useful to focus on very clear graphs and progressively introducing parts of the model.  I'll admit that the "Retriever" framing felt very confusing as well, as this models "Retrievers" don't seem to do any retrieving.
- For understanding the model, the "three world language" framing seems quite separate from the meat of what this model seems to be doing.    It implies that the somewhat simple "template" setup they use is important, but it seems to be only formed to teach a model what triplet completion is, which is something that one would think would be easily addressed in fine-tuning; there is not experimental exploration showing the value of those prompts.
- I'll admit to feeling suspicious at the high performance here (even after ablating nearly everything, they outperform most models?).

**Questions:**

-Looking at the code, it was unclear whether the  model is calculating the metrics (MRR, Hits@1, Hits@10) by ranking solely within a small batch containing the correct answer, or actually predicting the highest ranked entity from a full space of candidates. Could the authors clarify which one is being done?
- Could the authors clarify what is removed in the second , "text" ablation reow

**Limitations:**

yes, a broader impacts section is included and no major limitations are missing.

---

> ### Author Rebuttal · Authors · 2024-08-06
>
> Thank you very much for your constructive and detailed comments. We appreciate the opportunity to provide further clarifications.
>
>
> ### Weaknesses:
>
> - **Why is the proposed model better than the prior works? It is better to illustate each modules step-by-step with clear figures. The name of "Retriever" is confusing: what it retreves?**
>
>     Thank you for your insightful comments. We believe the reasons for the improved performance are threefold: (1) the power of LLMs; (2) the additional text information, where methods leveraging text features generally perform better; (3) the proposed MKGL effectively retrieves text and KG information for the LLM.
>
>     We indeed have a more detailed implementation section, including a step-by-step algorithm, which is currently in Appendix C due to space limitations. If the paper could be accepted, we plan to reintroduce this content into the main paper in camera-ready version (with an additional page allowance). We are willing to reintroduce the step-by-step process (Figure 1) of MKGL: Firstly, we construct input instructions and tokenize them into IDs. For out-of-vocabulary KGL tokens, their embeddings will be retrieved by our context retriever. Specifically (Figure 2), we aggregate the text tokens of each entity as its embedding, then use this embedding as the entity feature for aggregating KG information as the final KGL token embedding. Finally, we assemble the (KGL and LLM) token embeddings as the input sequence and feed them to the LLM to obtain output hidden states.
>
>     The term "retriever" simply implies that it can retrieve information from external resources that are helpful for inference. Since the KG information is not originally included in the input triplet, we designed a module to retrieve and process the information for the LLM. We have updated the introduction section in the revision to include an explanation for the name "retriever". Thank you again for your comment.
>
> - **The prompts used to fine-tune MKGL seems important, but it just teaches the model what triplet completion is. This can be easily addressed in fine-tuning.**
>
>     We also agree that the prompt templates may not be essential for fine-tuning an LLM on KG completion. In fact, the presentation of Instruction 3.1 in the main paper is not to demonstrate its novelty and importance. We simply aim to show how the input instruction is structured. Although we organize the context as a table, using sentences or other formats would not cause a significant performance loss.
>
>
> - **Lacking an explanation of the high performance (even after ablating nearly everything, they outperform most models?).**
>
>     Thank you for the comment. We do not ablate everything; the text retriever is actually replaced with a randomly initialized embedding module, as stated in Lines 285-287. We have highlighted this point in the caption of Table 4 and discussed the reasons in the revision. It is still a supervised fine-tuning LLM-based model, and it is expected to outperform most (not all) conventional models.
>
> ### Questions:
>
> - **It was unclear whether the model is calculating the metrics (MRR, Hits@1, Hits@10) by ranking solely within a small batch containing the correct answer, or a full space of candidates. Could the authors clarify which one is being done from the source code?**
>
>     Thank you for your detailed comments. To incorporate LLMs into KG completion, we developed a new framework based on the Transformers package provided by Hugging Face. The training and evaluation procedures also follow the corresponding suggestions. Although we have provided extensive comments in the uploaded code, it may not be as familiar as previous KG completion source code to the reviewer.
>
>     We do rank the target entities against all candidates, and we have highlighted this point in our paper in the revision. In the "predict" function of "llm.py" (Lines 299-325), we construct the candidate list according to the status of "self.training".
>
>     ```python
>     all_index = torch.arange(graph.num_node, device=device)
>     if self.training:
>         # train, do negative sampling
>         neg_index = self._strict_negative(
>          pos_h_index, pos_t_index, pos_r_index)
>         ...
>     else:
>         # test, constrcut testing examples (h,r,?) and (?,r,t)
>         h_index, t_index = torch.meshgrid(pos_h_index, all_index)
>         it_index, ih_index = torch.meshgrid(pos_t_index, all_index)
>         ...
>     ```
>
>     To ensure the evaluation function is correct, we have also tested the performance of a classical KG completion method, TuckER, within our framework, which achieves similar performance compared to its paper. We cannot provide an external link in the rebuttal, but TuckER is a very simple model. The reviewer may simply replace our model with its model (only 40 lines) and keep everything else unchanged to reproduce its results. Additionally, we adopt a strict ranking strategy where the target will be ranked below the entities with the same probability (Lines 278-286 in llm.py).
>
>
> - **Could the authors clarify what is removed in the second "Text" in the ablation results (Table 4)**
>
>     As stated in Line 286, toggling off "Text Retriever" does not mean we remove it, but we replace it with a learnable embedding module. This is equivalent to adding many new tokens into the LLM. The new tokens are randomly initialized and will be learned during fine-tuning. We have updated the caption of Table 4 to include this explanation.

---

> > ### Comment · Reviewer_dtBb · 2024-08-13
> >
> > I thank the authors for their detailed response! With the authors clarifications, I take back some of my initial concerns, and  I've revised my scores up from 4 to 6 .

---

> > > ### Author Response · Authors · 2024-08-13
> > >
> > > We sincerely appreciate your increased rating and recognition of the efforts we put into addressing your concerns! Your contribution to our work is highly valued and greatly appreciated.

---

### Official Review · Reviewer_4rXf · 2024-07-13

**Soundness:** 4
**Presentation:** 4
**Contribution:** 4
**Rating:** 8
**Confidence:** 4

**Summary:**

The authors introduce a SOTA method for allowing LLMs to incorporate information from knowledge graphs, relying on Knowledge Graph Language token embeddings to retrieve context, and then score it using a retriever that helps form a distribution over the possible entities to be incorporated.

**Strengths:**

Creative, interesting paper. Introduces a novel approach in a domain that could be of great use to data analysts (LLMs for knowledge graphs).
Well-supported empirically with strong evaluation on major benchmarks. Outperforms alternative methods mostly across the board.
Simple and effective illustrations. Good use of formatting in the paper itself to facilitate understanding (appreciated the use of color, especially).
Method may have notable benefits for reducing problems like hallucination and distribution drift, contributing to a solution for major outstanding issues with LLMs.

**Weaknesses:**

Could use a longer and more detailed discussion section; ends a little too abruptly.

**Questions:**

None.

**Limitations:**

I have no concerns about this paper being published.

---

> ### Author Rebuttal · Authors · 2024-08-06
>
> We are grateful for your encouraging comments and insightful suggestion. We hope the following response address your concern:
>
> ### Weaknesses:
>
> - **Could use a longer and more detailed discussion section; ends a little too abruptly.**
>
>     Thank you for your insightful suggestion. We also agree that a more detailed discussion could enhance understanding of our method and its implications across broader research areas. We have updated the conclusion section to include more comprehensive discussions on methodology, results, and potential value for other research fields:
>
>     The proposed context and score retrievers point out a new direction in incorporating LLMs with KGs for various tasks, such as question answering and entity linking. They also have implications in broader areas where the input cannot be precisely represented by text, e.g., node classification and protein representation learning. Furthermore, the construction of KGL vocabulary enables contrastive learning beyond tokens, offering insights into general machine learning. Hence, there are also numerous future directions. We plan to pretrain LLMs using a mixed corpus of KG and natural languages, enabling the LLM to comprehend and generate responses with linked data.

---

### Official Review · Reviewer_XXej · 2024-07-13

**Soundness:** 3
**Presentation:** 3
**Contribution:** 2
**Rating:** 7
**Confidence:** 2

**Summary:**

The paper proposes MKGL, a novel approach to integrate LLMs with KGs by instructing them in a specialized KG Language (KGL). KGL is a three-word language that mirrors the structure of KG triplets. The authors introduce a KGL context retriever and a score retriever, both based on LoRA, to efficiently encode textual and relational information into KGL token embeddings. MKGL outperforms existing KG completion methods, including LLM-based and conventional approaches, on both standard and inductive KG completion tasks. The paper also demonstrates MKGL's ability to generate valid KGL sentences and its computational efficiency compared to in-context learning methods.

**Strengths:**

* The authors present a novel approach to LLM-KG integration using a completion of entity-relation-entity triplets.
* The performance seems to be strong and the method outperforms the previous work on KG completion.
* At least compared to naive in-context learning, MKGL is more efficient and also achieves better scores.

**Weaknesses:**

* There are only limited details on certain aspects of the methodology, for example I couldn't find details about the actual implementation of the multi-layered PNA for KG information retrieval.
* While the authors claim that the proposed "three-word language" parsing of natural sentences is novel, it boils down to semantic-role labeling (SRL), a well-established NLP task. I believe that the paper should include a clear comparison to past SRL methods.
* The results in Table 2 should contain a column with computational cost (or at least the number of parameters of each method), to make it clear if it compares apples to apples.
* The computation runtime of the proposed method and the baselines is another thing that is lacking.

**Questions:**

* What exactly are the trainable parameters of the in-context-learning baseline in Section 4.6 and Figure 4? Isn't the point of ICL to not do any parameter updates at all and rely only on the contextual prompt?
* How does MKGL scale to much larger knowledge graphs? And how is it compared to other KG completion methods?

**Limitations:**

No issues found.

---

> ### Author Rebuttal · Authors · 2024-08-06
>
> Thank you very much for your helpful feedback and constructive suggestions. We have carefully integrated them into our paper.
>
> ### Weaknesses:
>
> - **Include more details of the methodology, e.g., the implementation of the multi-layered PNA for KG information retrieval.**
>
>     Thank you for your suggestion. The implementation of the multi-layered PNA closely follows the original paper, with the key difference lying in the input feature. As this isn’t a core contribution of MKGL, we have relocated it to Appendix B. In our revised version, we have elaborated on the encoding of KG information. Recall Equation (12) where the original multi-layer PNA can be written as:
>
>     $a_v^{(l+1)} = \delta \left( \bigoplus_{\rho \in \mathcal{P}} AGG_{\rho}(\{h_u^{(l)}, \forall u \in \mathcal{N}(v)\}), W^{(l)} \right)$
>
>     Here, $a_v^{(l+1)}$ represents the aggregated information for node $v$ at layer $l+1$, $\mathcal{N}(v)$ denotes the set of neighbors of $v$, $h_u^{(l)}$ signifies the hidden state of neighbor node $u$ at layer $l$, $\bigoplus$ is a concatenation operator, $AGG_{\rho}$ is an aggregation function (e.g., sum, mean, max), $\delta$ is a nonlinear activation function.
>
>     For knowledge graphs, there exists a relation (or edge type) $r$  between $u$ and $v$, which also needs to be encoded into the hidden states. To address this, we modify the above equation as:
>
>     $a_v^{(l+1)} = \delta \left( \bigoplus_{\rho \in \mathcal{P}} AGG_{\rho}(\{h_u^{(l)}\otimes h_r^{(l)}, \forall (v,r,u) \in \mathcal{T}\}), W^{(l)} \right),$
>
>     where $h_r^{(l)}$ denotes the hidden state of relation $r$ at layer $l$, and $\mathcal{T}$ denotes the triplet set. $\otimes$ is the operator used to combine the relation $r$ and neighboring node $u$, typically set as point-wise multiplication. Importantly, the hidden states at the first layer are not randomly initialized. They are the output of the text information retrieval, as illustrated in Figure 2c.
>
>
> - **Include a clear comparison to semantic-role labeling (SRL) methods.**
>
>     We have revised the related work section to compare KG completion with Semantic Role Labeling (SRL). Both tasks can be viewed as classification problems. However, the label spaces differ significantly. Most NLP tasks involve a smaller number of classes, usually less than 1,000, whereas for KG completion, the label space can exceed the vocabulary of LLMs. For example, the WN18RR dataset contains over 40,000 different entities, making it impractical to simply feed them all as possible results and let the LLM select one as output.
>
> - **Better to add a column with computational cost/parameter in Table 2.**
>
>     Many thanks. The number of parameters for MKGL is evidently greater than most conventional methods, a limitations we have discussed in the paper. A more comprehensive comparision metric may be the number of trainable parameters. Below are the results for FB15k-237 (the full table is available in the newly uploaded rebuttal.pdf), with some results sourced from [1], and others (marked by *) evaluated using the official repositories.
>
>     | Methods  | # FB15k-237 Trainable Parameters (M) | FB15k-237 MRR | # WN18RR Trainable Parameters (M) | WN18RR MRR |
>     |---|:---:|:---:|:---:|:---:|
>     | TransE  | 2 | .310 | 21 | .232 |
>     | RotatE  | 15 | .338 | 20 | .476 |
>     | TuckER  | 11* | .358 | 9* | .470 |
>     | CompGCN |  10* | .355 |  12* | .479 |
>     | CoKE | 10 | .364 | 17 | .484 |
>     | KG-BERT | 110* | - | 110* | .216 |
>     | StAR | 355* | .296 | 355* | .401 |
>     | KGLM | 355* | .289 | 355* | .467 |
>     | DET | 16 | .376 | 24 | .507 |
>     | **MKGL** | 20 | .415 | 20 | .552 |
>
>     It is evident that the number of trainable parameters for MKGL is comparable to that of conventional methods, and this gap narrows as the knowledge graph gets larger (WN18RR). Some language-model-based methods (e.g., KG-BERT) leverage a full-parameter-fine-tuning strategy, employing significantly more trainable parameters.
>
> - **The runtime of the proposed method and the baselines is not included.**
>
>    The runtime of LLM-based and conventional methods may be incomparable, but we compare our method with a vanilla supervised fine-tuning LLM-based method in Figure 4. The results demonstrate the high efficiency of our method.
>
> ### Questions:
>
> - **The name of ICL (1-hop)/ICL (2-hop) in Section 4.6 and Figure 4 may be confusing.**
>
>     Sorry for the confusion. The current names are inappropriate and misleading. Raw/ICL (1-hop)/ICL (2-hop) are methods involving new random-initialized token embeddings and score layers to represent every entity. In essence, each entity is considered a new token for the LLM, and is added to the vocabulary (accomplished using the native “resize_token_embeddings” function to add new tokens to LLM). The new embeddings do require training.
>
>     This experiment aims to verify whether the proposed text retriever and KG retriever are superior to initializing new tokens and directly incorporating the KG context information in the input, respectively. It’s crucial to emphasize the necessity of including such new token embeddings to estimate the probabilities of all entities. This explains why these variants have more trainable parameters than MKGL.
>
>     In the revised version (Figure 1 in rebuttal.pdf), we have renamed them and provided more detailed explanations: Raw has been renamed to NewToken, ICL (1-hop) is now NewToken (1-hop), and ICL (2-hop) is now NewToken(2-hop).
>
>
> - **How does MKGL scale to much larger knowledge graphs? And how is it compared to other KG completion methods?**
>
>     MKGL can easily scale to large KGs, as the number of trainable parameters remains independent of KG size. All KGL tokens stem from constant LLM token embeddings. This characteristic potentially positions MKGL as advantageous compared to conventional methods that initialize an embedding for each KG entity.
>
> [1] HittER: Hierarchical Transformers for Knowledge Graph Embeddings. EMNLP, 2021.

---

> > ### Comment · Reviewer_XXej · 2024-08-12
> >
> > Thank you very much for your response! I still believe that you should publish the (training & inference) computational cost and full parameter count, and compare it against the other methods. I don't see the significantly increased number of parameters as a negative, as long as it's transparent to the reader. Informing only about the training parameters seems somewhat misleading. I'm happy to increase the final score to 7 if you consider this.

---

> > > ### Author Response · Authors · 2024-08-12
> > >
> > > We are truly grateful for your increased rating and insightful comments. We completely agree with your point about listing the number of full parameters, which is important to provide a comprehensive comparison. While we are unable to revise "rebuttal.pdf" at this stage, we are fully committed to updating Table 2 to include the statistics of full parameters.
> > >
> > > In most methods, the number of trainable parameters aligns with the number of full parameters. However, there are two exceptions: KG-Llama and our method MKGL, both of which employ LoRA.
> > >
> > > | Methods | # Trainable Parameters (M) |# Full Parameters (M) |
> > > |---|:---:|:---:|
> > > | KG-Llama  | 13 | 6,755 |
> > > | **MKGL**  | 20 | 6,762 |
> > >
> > > MKGL incorporates additional neural layers for aggregating text and KG information, thus necessitating a greater parameter count. Once again, we sincerely thank you for your valuable suggestions.

---

> > > > ### Comment · Reviewer_XXej · 2024-08-12
> > > >
> > > > Thank you for the additional information!

---

> > > > > ### Author Response · Authors · 2024-08-13
> > > > >
> > > > > Thank you very much for your prompt feedback and unwavering support!

---

### Author Rebuttal · Authors · 2024-08-06

Dear all reviewers:

We sincerely appreciate the time and effort you have dedicated to reviewing our paper.

We would like to express our gratitude to Reviewers XXej for suggesting the inclusion of more details in the related work and methodology sections. We have incorporated these suggestions in the revision.

We are also grateful to Reviewers 4rXf for providing suggestions on the enrichment of the conclusion section. We have expanded our discussions not only limited to KG but also their potential impact on other areas.

We genuinely thank Reviewer dtBb for recommanding more details about the implementation and experimental settings. We have updated the corresponding paragraphs to cover more specific settings in the revision.

Furthermore, we extend our special thanks to Reviewer NCjt for pointing out the underlying limitations and providing  insightful solutions. We have added a standalone section to discuss the limitations and conducted experiments to analyze them.

Thanks again to all reviewers. Your comments are invaluable in helping us enhance the quality of our paper.

Best Regards,

Authors

---

### Decision · Program_Chairs · 2024-09-25

**Decision:**

Accept (spotlight)

**Comment:**

The authors introduce a method for LLMs to carry out knowledge graph completion.  The system outperforms a number of established baselines.  Reviewers agree that that the method is sound, novel and impactful, if only in applications.  No substantial weaknesses remain after the rebuttal/discussion periods.